# In Silico and In Vitro Analyses of Angiotensin-I Converting Enzyme Inhibitory and Antioxidant Activities of Enzymatic Protein Hydrolysates from Taiwan Mackerel (*Scomber australasicus*) Steaming Juice

**DOI:** 10.3390/foods11121785

**Published:** 2022-06-17

**Authors:** Fenny Crista A. Panjaitan, Ting-Yi Chen, Hao-Hsiang Ku, Yu-Wei Chang

**Affiliations:** 1Marine Products Processing Study Program, Marine and Fisheries Polytechnic of Jembrana, Bali 82218, Indonesia; fennycap@gmail.com; 2Department of Food Science, National Taiwan Ocean University, Keelung City 20224, Taiwan; grazychen@gmail.com; 3Institute of Food Safety and Risk Management, National Taiwan Ocean University, Keelung City 20224, Taiwan; kuhh@email.ntou.edu.tw

**Keywords:** mackerel steaming juice, protein hydrolysates, in silico, ACE-inhibitory, antioxidant

## Abstract

Mackerel (*Scomber australasicus*) steaming juice (MSJ) can be a good source of proteins. However, it is often treated as food waste during the canning process. The objective of this study was to investigate the Angiotensin-I converting enzyme (ACE-I) inhibitory and antioxidant activities from MSJ hydrolysates using in silico and in vitro approaches. Proteins extracted from MSJ were identified by proteomic techniques, followed by sulfate polyacrylamide gel electrophoresis (SDS-PAGE), in-gel digestion, tandem mass spectrometry and on-line Mascot database analysis. Myosin heavy chain (fast skeletal muscle), actin, myosin light chain 1 (skeletal muscle isoform), collagen alpha-2(I) chain, tropomyosin alpha-1 chain, beta-enolase, fructose-bisphosphate aldolase A and glyceraldehyde-3- phosphate dehydrogenase were identified and further analyzed using BIOPEP-UWM database. In silico results indicated that MSJ proteins had potential bioactive peptides of antioxidant and ACE-I inhibitory activities. MSJ was then hydrolyzed using six proteases (papain, pepsin, proteinase k, alcalase, bromelain, thermolysin). In particular, pepsin hydrolysates (5 mg/mL) showed the highest 2,2-diphenyl-1-picrylhydrazyl (DPPH) radical scavenging activity (61.54%) among others. Alcalase hydrolysates (5 mg/mL) exhibited the highest metal chelating activity (89.76%) and proteinase K hydrolysates (5 mg/mL) indicated the highest reducing power activity (1.52 abs). Moreover, pepsin hydrolysates (0.1 mg/mL) possessed the highest ACE inhibitory activity (86.15%). Current findings suggest that MSJ hydrolysates can be a potential material to produce ACE-I inhibitory and antioxidant peptides as nutraceutical or pharmaceutical ingredients/products with added values.

## 1. Introduction

Mackerel (*Scomber australasicus*) has contributed to the major production among all of the landings caught from offshore seawater, in Taiwan. In addition, around 80% of caught mackerel fish are made into canned products. In 2014, the total canned mackerel production in Taiwan was 8155 metric tons, accounting for 83.9% of the total production of canned fishery products [1].

Steaming is an important process in the canning processing line of fishery products. In this process, a large amount of steaming juice was produced. In average, one-ton mackerel can produce 140–150 kg of steaming juice. However, the steaming juice which contained high biological oxygen demand (BOD) value was usually considered as liquid waste during the process. Moreover, discharging steaming juice without proper treatment may lead to severe water pollution and to the loss of valuable protein [2]. Each year, mackerel canning process produced 1182 tons of steaming juices. Hence, the utilization of mackerel steaming juice has to be developed to recover soluble protein sources in canned fish industries and to improve their economic value.

Angiotensin-I converting enzyme (ACE-I) has been widely known to have an important role in the renin-angiotensin system that regulates blood pressure. It converts angiotensin-I (Ang-I) into angiotensin-II (Ang-II) and catalyzes the degradation of bradykinin. Ang-II is a vasoconstrictor and growth-promoting substance, whereas bradykinin is a potent vasodilator and growth inhibitor [3]. ACE-I inhibitors are very important for the treatment of hypertension to prevent the conversion type of angiotensin [4]. ACE-I inhibitors, such as Captopril, Enalapril, Alacepril, are generally used as the first-line therapy of hypertension. However, several side effects occurred while taking this kind of medicine, including cough, skin rashes, loss of taste, angioneurotic edema [5]. Therefore, many researches have been tried to examine the ACE-I inhibitory activity of food components or their hydrolysates to lower blood pressure without any further side effects. Many ACE-I inhibitory peptides have been isolated, some of which are able to lower the systolic blood pressure (SBP) of spontaneously hypertensive rats (SHR) [6,7,8]. Furthermore, antioxidant activity has been also reported for protein hydrolysates prepared from various fish sources such as capelin, mackerel, yellowfin sole, Alaska pollack, Atlantic salmon, Hoki, conger eel and scads [9]. Antioxidants have capability to slow cells damage induced by free radicals and to play an important role in human health and food preservation. Synthetic antioxidants, such as butylated-hydroxytoluene (BHT), butylated-hydroxy anisole (BHA), tertbutyl hydroquinone (TBHQ), and propyl gallate (PG), are generally used to remove free radicals and to slow down oxidation processes effectively. However, these synthetic antioxidants are used under strict regulation to restrict potential health hazards in the food and medicinal industries [10]. Therefore, bioactive peptides from food origins containing those activities could be alternative sources to treat hypertension and free radicals.

Bioactive peptides usually contain 3–20 amino acid residues, and their activities are based on their amino acid composition and sequence [11]. These physiologically active peptides can be obtained through different protein hydrolysis methods such as solvent extraction, enzymatic reaction, radiation, and microorganisms. In addition, enzymatic hydrolysis can be performed under mild conditions and could avoid the extreme conditions required by chemical treatments [11].

In recent years, many studies were conducted on protein hydrolysates in steaming juice. Findings indicated that salmon steaming juice was consisted of antioxidant, hypoglycemic, antihypertensive, anti-cancer, and other physiological activities [12,13,14,15,16,17,18]. In addition, mackerel steaming juice (MSJ) contained antibacterial, antioxidant, and other physiological activities [19,20,21,22]. However, there are insufficient studies on the active peptide related to MSJ. Therefore, objectives of this study were to determine protein contents in MSJ, to identify protein sequences in MSJ by proteomics techniques, to study potential bioactive peptides generated by enzymatic hydrolysis, as well as to investigate antihypertensive and antioxidant activities of MSJ.

## 2. Materials and Methods

### 2.1. Materials

Mackerel (*Scomber australasicus*) steaming juice was supplied by Tong Yeng Co., Ltd., Yilan county, Taiwan. Enzymes such as papain, pepsin, thermolysin, bromelain and alcalase were purchased from Sigma-Aldrich Co., Ltd. (St Louis, MO, USA). Proteinase K was purchased from Promega Corporation. ACE from rabbit lung (≥2 units/mg protein) and the substrate N-(3-[2-furyl]acryloyl)-phenylalanylglycylglycine (FAPGG) and DPPH (2,2-diphenyl-1 picrylhydrazyl were purchased from Sigma Aldrich. Other chemicals and reagents used in this research were of analytical grade.

### 2.2. Proximate Analysis of Mackerel Steaming Juice (MSJ)

Moisture, ash, crude protein and crude fat content of MSJ were determined according to official methods adopted by Association of Official Analytical Chemists [23].

### 2.3. Determination of pH and Salinity Value

The pH value of 25 mL MSJ was measured by using pH meter (CyberScan pH 510) at room temperature. Meanwhile, salinity was measured by using titration method modified from Chinese National Standard CNS 423 N 5006. An aliquot of 5 mL sample solution (W) is made up to 50 mL with deionized water. Then, 1 mL of 2% potassium chromate (K_2_CrO_4_) solution was used as an indicator and added to 5 mL of the solution. Titration (V mL) was carried out using 0.02 N silver nitrate and stopped when mild orange color formed. NaCl content was calculated as follows:$$NaCl\;\left(\%\right)=0.02\times V\times 0.00117\times \frac{50}{5}\times \frac{1}{W}\times 100\%\;$$

### 2.4. Preparation of MSJ Powder

By referring to the method of Ko et al. [18], MSJ was sealed in a vacuum bag and stored overnight in the refrigerator at 4 °C. The oil floated on top of the MSJ was removed by filtration and the mixture was centrifuged at 5000× *g* for 15 min (4 °C). Then, the supernatant was freeze-dried and stored at −20 °C.

### 2.5. Preparation of Protein Isolates

Freeze-dried MSJ (0.5 g) and muscle of mackerel (*Scomber australasicus*) (0.5 g) were homogenized, respectively, with ten times volumes of sodium dodecyl sulfate (SDS) Tris(hydroxymethyl)aminomethane buffer solution (10 mM Tris-HCl, 5 mM PMSF, 2% SDS, pH 7.2) for 2 min at 6 m/s using a high-speed benchtop homogenizer (FastPrep^®^-24) [24]. Then, the mixture was heated in a boiling water bath (100 °C) for 5 min. After heating, sample was re-homogenized for 2 min and centrifuged (13,500 rpm, 20 min). Supernatant was collected for the following test.

### 2.6. Determination of Protein Content, Yield and Protein Recovery

Protein content of Mackerel Steaming Juice (MSJ) powder and MSJ protein isolates (SDS extracts) was determined by Kjeldahl nitrogen method [23]. The weight of MSJ (W_1_) and freeze-dried MSJ powder (W_2_) were measured, respectively. The protein yield and protein recovery were calculated as follows:$$Protein\;yield\;\left(\%\right)=\frac{W_{2}}{W_{1}}\times 100\%\;$$
Protein recovery (%)=Protein Content×Extraction Yield×100%

### 2.7. Sodium Dodecyl Sulphate-Polyacrylamide Gel Electrophoresis (SDS-PAGE)

Molecular mass distributions of protein were analyzed by using SDS-PAGE with 4% stacking gel (*w/v*) and 12% polyacrylamide gel (*w/v*) as modified from Huang et al. [25]. Sample (1.5 mg) was dissolved in 100 μL of sample buffer (ddH_2_0 3.55 mL, 0.5 M Tris-HCl pH 6.8 1.25 mL, Glycerol 2.5 mL,10% SDS 2 mL, 0.5% Bromophenol blue 0.2 mL, 2-Mercaptoethanol 0.5 mL) and heated at 95 °C for 5 min. After being centrifuged at 4000 rpm for 5 min, the supernatant was taken as a sample solution. Each sample was loaded into the well. Then, the electrophoresis was run at 70 V for 3 h. Afterwards, the gel was stained with Brilliant Blue (Bio-Rad, Coomassie R250) for 30 min, and destained using a destaining solution (water: methanol: acetic acid; 7:2:1, *v/v/v*).

### 2.8. Proteomics Technique

Proteomic techniques were adapted from Huang et al. [25] and Ostasiewicz et al. [26]. Gel with intensive bands was placed on a clean glass slide and each band was cut into pieces (1 mm^3^). Next, gel pieces were transferred into microcentrifuge tube and destained with 200 μL of 50% acetonitrile (ACN)/25 mM ammonium bicarbonate (ABC). Then, 100-μL of 50 mM dithioerythritol (DTE)/25 mM ABC was added for 60 min at 37 °C and discarded. The procedure was followed by the addition of 100 μL iodoacetamide (IAM)/25 mM ABC and incubated for 60 min in the dark at room temperature. Excess solutions in the microcentrifuge tube were removed. The remaining gels were then washed with 200 μL of 50% ACN/25 mM ABC for 15 min. This process was repeated four times. After the solution was fully removed, 100% ACN was added into gels for 5 min and dried for about 5 min using Speed Vac (Thermo Scientific, Waltham, MA, USA). After that, gels were proceeded with the addition of Lys-C protease solution (enzyme: protein, 1:50) and Trypsin for 16 h at 37 °C. Tryptic peptides were then extracted with 50 μL of 50% ACN/5% trifluoroacetic acid (TFA). The final extraction was transferred into new microcentrifuge tube and dried with Speed Vac device. Tryptic peptides were then purified with zip-tip purification process; the sample peptides were stored for further UPLC/Q-TOF-MS/MS analysis.

### 2.9. BIOPEP-UWM Database Analysis of Bioactive Peptides and Enzyme Cleavages

According to Huang et al. [25], proteins sequences identified from UPLC/Q-TOF-MS/MS analysis were examined using BIOPEP-UWM database (https://biochemia.uwm.edu.pl/en/biopep-uwm-2/, accessed on 28 March 2022) [27] to predict biological activities and to simulate proteolytic hydrolysis. Firstly, “Bioactive peptides” was chosen and the “profiles of potential biological activity” option was selected to investigate bioactive peptides potentially generated from identified protein sequences. Other information such as BIOPEP ID, name of peptides, biological activity, quantity of peptides in the protein sequence, peptide sequence and position were also revealed. Thereafter, the simulation of enzymatic hydrolysis was carried out by choosing the “enzyme(s) action” tool in the database.

### 2.10. Preparation of Enzymatic Hydrolysate

Enzymatic hydrolysis was conducted by referring the method of Huang et al. [25] with modification. One gram (protein basis) of Mackerel Steaming Juice (MSJ) powder was suspended into 100 mL deionized water; 10 mg of enzyme powder was added at the optimum temperature and pH of the enzyme (Papain pH 7, 55 °C; Pepsin pH 2, 37 °C; Thermolysin pH 8, 70 °C; Proteinase K pH 8 50 °C; Bromelain pH 7, 50 °C; Alcalase pH 8, 60 °C). The hydrolysis time was fixed at 4 h and the pH value of the mixture were monitored at every 30 min. Then, the enzyme was inactivated by heating the mixture at 100 °C for 10 min. The mixture was then cooled to 4 °C and centrifuged at 8000 rpm for 20 min. The hydrolysate was lyophilized and stored at 20 °C for further analysis.

### 2.11. Degree of Hydrolysis (DH)

The o-phthalaldehyde (OPA) method was used to determine the degree of hydrolysis. According to Charoenphun et al. [28], an aliquot of hydrolysates (10 µL) and Gly-Gly-Gly standard (5 µL) were mixed with 200 µL of fresh OPA solution (100 mM sodium tetraborate 12.5 mL, 20% SDS 1.25 mL, o-Phthalaldehyde 20 mg/0.5 mL methanol, 2-Mercaptoethanol 0.05 mL, ddH_2_O 10.7 mL). The mixture was then incubated for 100 s at 37 °C. Total acid hydrolysis was performed by adding 6 N HCl into hydrolysates, stirred for 24 h at 100 °C prior to analysis. The absorbance was measured at 340 nm using multiple reader (Multiskan Go, Thermo Fisher Scientific, Waltham, MA, USA). The degree of hydrolysis (%) were calculated using the following formula:$$DH\left(\%\right)=\left[\frac{{\left(NH_{2}\right)}_{tx}-{\left(NH_{2}\right)}_{t0}}{{\left(NH_{2}\right)}_{total}-{\left(NH_{2}\right)}_{t0}}\right]\times 100\%\;$$
where, (NH_2_)_tx_ is the amount of free amino groups at X min (mg/mL); and (NH_2_)_total_ is the amount of total amino groups by total acid hydrolysis (mg/mL). (NH_2_)t_0_ represents the amount of free amino groups at 0 min of hydrolysis (mg/mL).

### 2.12. 1,1-Diphenyl-2-picrylhydrazyl (DPPH) Radical Scavenging Assay

DPPH radical assay was used to determine antioxidant properties of protein hydrolysates according to Girgih et al. [29]. A 100 µL aliquot of hydrolysates in methanol (5 mg/mL) was mixed with 0.1 mM DPPH methanolic solution. The mixture was incubated for 30 min in the dark. The absorbance was measured using a microplate reader (Multiskan Go, Thermo Fisher Scientific, Waltham, MA, USA) at 517 nm against blank samples. Methanol and ascorbic acid were used as negative and positive control, respectively. DPPH radical scavenging ability was calculated with the following equation:$$DPPH\;radical\;scavenging\;activity\;\left(\%\right)=\left(\frac{OD517control-OD517sample}{OD517control}\right)\times 100\%\;$$

### 2.13. Determination of Fe^2+^ Chelating Activity

Metal chelating activity was measured based on Girgih et al. [29]. A 200 µL aliquot of hydrolysates with 2 mM FeCl_2_ µL and 5 mM Ferrozine were mixed for 10 min in the dark. The absorbance was measured at 562 nm against blank samples using a microplate reader (Multiskan Go, Thermo Fisher Scientific, Waltham, MA, USA). Ethylenediaminetetraacetic acid (EDTA) was used as positive control.
$$Metal\;chelating\;activity\;\left(\%\right)=\left(\frac{OD562control-OD562sample}{OD562control}\right)\times 100\%\;$$

### 2.14. Reducing Power Assay

Reducing power of MSJ and MSJ hydrolysates was determined according to Girgih et al. [29]. A 250 µL aliquot of hydrolysates (5 mg/mL) was dissolved in 250 µL of 0.2 M sodium phosphate buffer (pH 6.6) (1 mg/mL) and mixed with 250 µL of 1% (*w/v*) potassium ferricyanide solution followed by incubation for 20 min at 50 °C. Afterwards, 10% (*w/v*) TCA (250 µL) were added into the mixture and centrifuged at 3000 rpm for 5 min. The collected supernatant (500 µL) was mixed with 500 µL of distilled water and 0.1% (*w/v*) ferric chloride in dark for 10 min. Then, the absorbance of 200 µL of mixture was measured at 700 nm using a microplate reader (Multiskan Go, Thermo Fisher Scientific, Waltham, MA, USA). Distilled water was used as negative control; ascorbic acid as positive control. The increasing of reducing power activity is in accordance to the increasing of absorbance.

### 2.15. Angiotensin-I Converting Enzyme (ACE-I) Inhibitory Assay

The ability of protein hydrolysates to inhibit the activity of ACE-I was determined by the combination method of Pihlanto et al. [30] and Udenigwe et al. [31]. A 0.1 mg of samples were dissolved in 50 mM Tris-HCl buffer containing 0.3M NaCl at pH 7.5. Then, 20 µL mixture and 10 µL ACE (0.5 U/mL final activity of 25 mU) were added to 170 µL of 0.5 mM N-[3-(2-Furyl) acryloyl]-L-phenylalanyl-glycyl-glycine (FAPGG)/50 mM Tris-HCl. The rate of decrease in absorbance at 345 nm was monitored at regular intervals for 30 min at 37 °C in a microplate reader (Multiskan Go, Thermo Fisher Scientific, Waltham, MA, USA) at 340 nm (OD340). Tris-HCL buffer was used as the negative control. ACE activity is expressed as the rate of reaction (ΔA/min) and the inhibitory activity was calculated as follows:$$\mathrm{ACE-I}\;\mathrm{inhibition}\;(\%)=[1-\Delta {\mathrm{Amin}}^{-1}(\mathrm{sample})/\Delta {\mathrm{Amin}}^{-1}(\mathrm{control})]\times 100\%$$

### 2.16. Statistical Analysis

Statistical analysis was performed using SPSS (Statistical Package for Social Science) version 25.0. (SPSS, Chicago, IL, USA). One-way analysis of variance (ANOVA) was used in this study. The significant differences between mean values for the tests were determined by *t*-test or Ducan (Multiple Range Test).

## 3. Results and Discussion

### 3.1. Proximate Composition, pH and Salinity Value of Mackerel Steaming Juice (MSJ)

Proximate analysis of mackerel steaming juice (MSJ) collected from Tong Yeng company was measured. MSJ showed high percentage of moisture (93.76%), followed by crude protein (5.29%), ash (0.84%) and low percentage of crude fat (0.19%), as shown in Table 1. Results revealed that MSJ is composed of considerably moisture content. Those findings were in-line with the research reported by Chuo [32] showing that cooking juice samples of mackerel had 94.96–96.35% of moisture. It was also reported that protein content of MSJ was higher than sardine stickwater (4.53%) and tuna cooking water (3.86%) [33]. In addition, dry matter of MSJ allowed the protein content to be increased up to 84.94% indicating that protein is the main component of MSJ. Owing to rich content of protein, MSJ is potential sources of protein and bioactive peptides.

NaCl content and pH of MSJ were also observed at 0.38% and 5.93, respectively. According to Sun et al. [34], pH value can be used as an indicator of fish freshness. It was reported that migratory fish has a pH value of 5.6 to 6.0. Therefore, MSJ was considered fresh as the pH value was still in range. Moreover, the presence of salt in MSJ might be part of ingredients added during the canning processing. Corresponding to that, MSJ could be considered to have water-soluble and salt-soluble proteins.

### 3.2. Identification of Protein in Mackerel Steaming Juice (MSJ)

MSJ was defatted and freeze-dried into fine powder. Thereafter, some of MSJ powder (defatted sample) were subjected to protein extraction using sodium dodecyl sulphate (SDS) to gain SDS-soluble protein fraction. Results showed that crude protein of defatted MSJ was 89.72 ± 1.12% which was higher than that of MSJ (84.94%) in dry matter. This finding revealed that defatting process could increase protein content of food product. Furthermore, protein content of MSJ SDS-extract was reported at 75.72 ± 1.68% with yield and protein recovery were 75.39% and 57.09%, respectively. According to results, total SDS-soluble proteins was lower than defatted MSJ before extraction. This might be influenced by the extraction process causing the loss of protein or the forming of insoluble protein fraction due to denaturation and aggregation.

The molecular characteristic of MSJ protein was also observed. Figure 1 depicted the distribution of MSJ protein on SDS-PAGE based on the molecular weight with protein marker ranging from 17 to 180 kDa. Protein of defatted MSJ (Lane A) was distributed and detected mostly above 63 kDa, while the concentration of protein bands obtained from MSJ SDS extract (Lane B) showed a thick band around 35 kDa and gradually decreased above 130 kDa. Moreover, proteins of mackerel muscle (Lane C) were presented in some bold bands distributed evenly on gel. Several protein bands were marked with rectangle boxes and labelled as A1–A6, B1 and C1–C4. They were then sent for proteomics analysis, including in-gel digestion and UPLC/Q-TOF-MS/MS analysis to acquire the protein hits detected in each band, as shown in Appendix A. Those tryptic peptides were determined to have doubly and triply charged signals. Figure 2 depicted the representative of LC-MS/MS chromatogram of IEDEQSLGAQLQK (MW 729.88) identified from MSJ in band A4. Insert (1) shows the double charge signal from the 0.5 difference between adjacent signals, while insert (2) displays the fragmentation of LC MS/MS spectrum of the identified tryptic peptides.

Corresponding tryptic peptides of MSJ proteins from band A4 identified through LC-MS/MS analysis was furtherly detected in myosin heavy chain (fast skeletal muscle) sequence, as completely depicted in Appendix A. Sequences of tryptic peptides from myosin heavy chain (fast skeletal muscle, SwissProt: Q90339) matched to MSJ band A4 using MS/MS Ions search were marked in red letters as illustrated in Figure 3. Red letters found in protein sequences were tryptic fraction detected in sample during analysis. Trypsin was common enzyme used to cleave protein in proteomics approach [35]. Food proteins identified from Mascot database was summarized in Table 2. Proteins from those excised gels were observed including the protein hits, molecular weight (kDa) from NCBI database, the estimation of molecular weight from selected gels on SDS-PAGE. Findings showed that several proteins were not detected in MSJ samples, such as actin, alpha skeletal muscle B; action, cytoplasmic 2; actin, aplha anomalous and actin, cytoplasmic 3. In addition, collagen alpha-2(I) chain was not detected both in MSJ SDS and muscle SDS.

### 3.3. BIOPEP-UWM Databse Analysis of Mackerel Steaming Juice (MSJ)

The analysis of BIOPEP-UWM database was conducted to reveal potential bioactive peptides derived from MSJ proteins. Eight proteins (collagen alpha-2(I) chain, myosin heavy chain (fast skeletal muscle), actin (alpha skeletal muscle), tropomyosin alpha-1 chain, beta-enolase, fructose-biophosphate aldolase, glyceraldehyde-3-phosphatedehydrogenase and myosin light chain-1 (skeletal muscle isoform)) were chosen for the estimation of biological activity using the “profiles of potential biological activity” tool. Table 3 listed potential bioactivities generated from protein identified in MSJ. Common bioactive peptides detected in each protein were dipeptidyl peptidase IV inhibitor (DPP-IV), ACE-I inhibitor, antithrombotic, antioxidative and antiamnestic. Results showed that DPP-IV inhibitory peptides were dominant bioactive peptides possessed in MSJ proteins, followed by ACE inhibitors. Figure 4 showed bioactive peptides profilling of actin (SwissProt: P49055) identified from MSJ. In silico analysis revealed that actin was majorly composed of DPP-IV inhibitor (marked with yellow lines), ACE inhibitor (marked with blue lines) and antioxidative (marked with red lines) peptides. Abundant DPP-IV and ACE inhibitory peptides could be affected by structure and composition of amino acid. Moreover, bioactive peptides was presented in dipeptides and tripeptides with some overlapping activities. Davy et al. [36] and Wu et al. [37] mentioned that hydrophobic amino acids have important roles in possessing DPP-IV and ACE inhibitory activities.

The simulation of proteolysis was also conducted using BIOPEP-UWM database through “enzyme action” tool. Nine different enzymes were selected to simulate hydrolysis, such as ficain, papain, pepsin (pH > 2), chymotrypsin C, proteinase K, thermolysin, pancreatic elastase, cathepsin G and bromelain. Table 4 summarized the number of potential bioactive peptides released from MSJ proteins. Results showed that DPP-IV inhibitory peptides were largely generated from MSJ proteins, followed by ACE-I inhibitory peptides which was in-line with Table 3.

### 3.4. Peptide Profile of MSJ Protein Hydrolysates

Several proteases were furtherly chosen for in vitro hydrolysis, such as papain, pepsin, bromelain, thermolysin, alcalase, and proteinase K for 4 h under optimal conditions. Degree of hydrolysis (DH) from each protease was shown in Figure 5. Based on results, DH of those six enzymes ranged between 5–30%. MSJ protein hydrolysates showed relatively high DH at the first 30 min of hydrolysis and then getting slower thereafter. This finding indicated that the cleavage of peptides was significantly occurred during the initial 30 min, then the ratio of enzyme and substrate was out of equilibrium after 4 h [38]. The same result was also observed from tuna cooking juice hydrolysis using various proteases [13,38,39,40].

After 4 h of hydrolysis, proteinase K was observed to have the highest degree of hydrolysis (26.18%), followed by alcalase (10.12%), thermolysin (9.88%), bromelain (8.38%), papain (5.69%) and pepsin (4.64%). As shown in Figure 6, proteinase K revealed significant differences (*p* < 0.05) among other proteases.

Peptide contents and yield of each hydrolysate were determined. As summarized in Table 5, MSJ proteinase K hydrolysates served the highest peptide content (326.92 mg/g), followed by MSJ alcalase hydrolysates (260.05 mg/g). Results was in correspondence with Figure 6 showing that high percentage of DH produced high peptide content. However, MSJ pepsin hydrolysates generated the lowest peptide content at 124.00 mg/g, which were lower than MSJ pre-hydrolysis at 162.14 mg/g. Moreover, MSJ proteinase K hydrolysates also yielded the highest percentage of protein recovery at 98.20%, followed by MSJ pepsin hydrolysates and MSJ thermolysin hydrolysates at 92.79%.

Patterns of proteins from each hydrolysate were analyzed through ten percent of SDS-PAGE with the MW distribution of protein standard ranged from 17 to 180 kDa, as depicted in Figure 7. Results showed the transformation of MSJ protein from large MW into small MW of hydrolysates below 48 kDa, while the MW distribution of proteinase K hydrolysates was not detected on the gel due to high DH resulted in low molecular weight below 17 kDa. Result proved that hydrolysis cleaved proteins into small peptides with low molecular weights [41].

### 3.5. Antioxidant Properties

Antioxidant activities, 2,2-diphenyl-1-picrylhydrazyl (DPPH) radical scavenging activity, metal chelating activity and reducing power, of MSJ and MSJ hydrolysates were determined accordingly. DPPH is stable free radicals which commonly used for the analysis of antioxidant activity [13]. Figure 8A revealed the DPPH radical scavenging activity of MSJ (5 mg/mL), MSJ hydrolysates (5 mg/mL) and vitamin C (0.05 mg/mL) showing that pepsin could possess the best DPPH radical scavenger at 61.65% (*p* < 0.05), while other hydrolysates were not significantly different from MSJ protein (29.1%) (*p* > 0.05). In addition, DPPH radical scavenging activity was not detected on papain and bromelain hydrolysates. Size, level and composition of amino acids affect the antioxidant activity [22]. Furthermore, pepsin hydrolysate was then furtherly analyzed with different concentration as shown in Figure 8B. The increasing of hydrolysates concentration produced better DPPH radical scavenging activity. However, pepsin hydrolysates at 2.5 mg/mL did not significantly possess better activity than those at 3.5 mg/mL and 5 mg/mL (*p* > 0.05) showing that protein concentration of 2.5 mg/mL, 3.5 mg/mL and 5 mg/mL scavenged DPPH radicals in the same capacity.

Fe^2+^ chelating activity of MSJ, MSJ hydrolysates and EDTA, at 5 mg/mL accordingly, were illustrated in Figure 9A. Corresponding to results, protein hydrolysates of MSJ generated significantly better Fe^2+^ chelating activity than those of MSJ (*p* < 0.05). Moreover, alcalase hydrolysates exhibited the highest metal chelating activity among other hydrolysates, at 89.76%, which was not significantly different (*p* > 0.05) from those of thermolysin and EDTA, at 87.38% and 93.35%, respectively. Alcalase hydrolysates were then treated with dose dependent manner as shown in Figure 9B. Results showed that the Fe^2+^ chelating activity raised as the increasing of hydrolysates concentration. However, activity of sample at 2.5 mg/mL was not significantly different from those of 3.5 and 5 mg/mL (*p* < 0.05). In other words, alcalase hydrolysates at 2.5 mg/mL could generate similar Fe^2+^ chelating activity to those at 3.5 and 5 mg/mL.

Reducing power of MSJ (5 mg/mL), MSJ hydrolysates (5 mg/mL) and vitamin C (0.05 mg/mL) was investigated as shown in Figure 10A. Proteinase K hydrolysates generated the highest reducing power (1.52), followed by bromelain (1.38), thermolysin (1.25) and alcalase (1.24), while pepsin had the lowest reducing power (0.414) which was lower than MSJ (0.63). Furthermore, reducing power of Proteinase K hydrolysates at different concentration were illustrated in Figure 10B. Significant differences were observed with the concentration started from 0.005 to 3 (*p* < 0.05), where the reducing power in this interval increased as the increasing of hydrolysates concentration. In addition, proteinase K hydrolysates at 3 mg/mL and 5 mg/mL revealed no significant differences (*p* > 0.05).

Based on the experimental results, this study found that proteases exhibited higher degree of hydrolysis, such as proteinase K, alcalase and thermolysin, produced better Fe^2+^ chelating activity and reducing power. Conversely, pepsin hydrolysates with the lowest hydrolysis rate revealed relatively low Fe^2+^ chelating activity and reducing power, but significant DPPH radical scavenging activity compared to other hydrolysates. Antioxidant activity of protein hydrolysates depends on the protease and conditions employed during hydrolysis [13]. Moreover, a previous study conducted by Wu et al. [22] reported that mackerel hydrolysates were capable to possess DPPH free radical quenching activity and reducing power which might be related to the presence of carnosine and anserine within the peptide sequences.

### 3.6. Angiotensin Converting Enzyme I (ACE-I) Inhibitory Activity

Angiotensin-I-Converting enzyme (ACE-I) is the enzyme used to regulate blood pressure. Hence, ACE inhibitory activity can be used as an index to determine the antihypertensive effect. Some studies related to ACE-I inhibitory activity derived from seafood cooking or steaming juice have been studied; for instance, tuna cooking juice hydrolyzed by spleen enzymes showed good activity of ACE-I inhibitory (58.5 ± 0.13%, at DH 80.5 ± 3.12%) [38]. The analysis ACE-I inhibitory activity derived from MSJ and MSJ hydrolysates was displayed in Figure 11. Based on the result, ACE-I inhibitory activity was optimally exhibited after hydrolysis (*p* < 0.05). Corresponding to this finding, proteolysis could improve ACE-I inhibitory activity through the formation of shorter chain peptides, depending on enzyme activities. In addition, short fraction of peptides contributed to the increasing of ACE-I inhibitory activity, as previous studied by Kasiwut et al. [38] in tuna cooking juice hydrolysates. Pepsin hydrolysates possessed the highest ACE-I inhibitory activity (86.14%) (*p* < 0.05), while ACE-I inhibitory activity of other five hydrolysates were not significantly different (*p* > 0.05) among them. This finding was in compliance with the identification of BIOPEP database indicated that pepsin hydrolysates produced the highest amount of ACE-I inhibitory peptides.

## 4. Conclusions

Eight proteins were identified in mackerel steaming juice (MSJ) through proteomics analysis. In silico study showed that mackerel steaming juice (MSJ) is potential to become alternative sources of bioactivities through BIOPEP-UWM database. Results showed that MSJ potentially generated dipeptidyl peptidase IV (DPP-IV) inhibitor, angiotensin converting enzyme (ACE) inhibitor, antithrombotic, antioxidative and antiamnestic peptides. In vitro analysis was then proved to determine antioxidant and ACE-I inhibitory activities. Pepsin possessed the best DPPH radical scavenger and ACE-I inhibitory activity, at 61.65% and 86.14%, respectively; alcalase hydrolysates exhibited the highest metal chelating activity at 89.76%; proteinase K generated the highest reducing power at 1.52. According to results, pepsin is in accordance to in silico results to produce better ACE-I inhibitory activity. Owing to these findings, recovered proteins of MSJ were recommended to be nutraceutical ingredients or drug development potential with ACE-I inhibitory and antioxidant activities.

## Figures and Tables

**Figure 1 foods-11-01785-f001:**
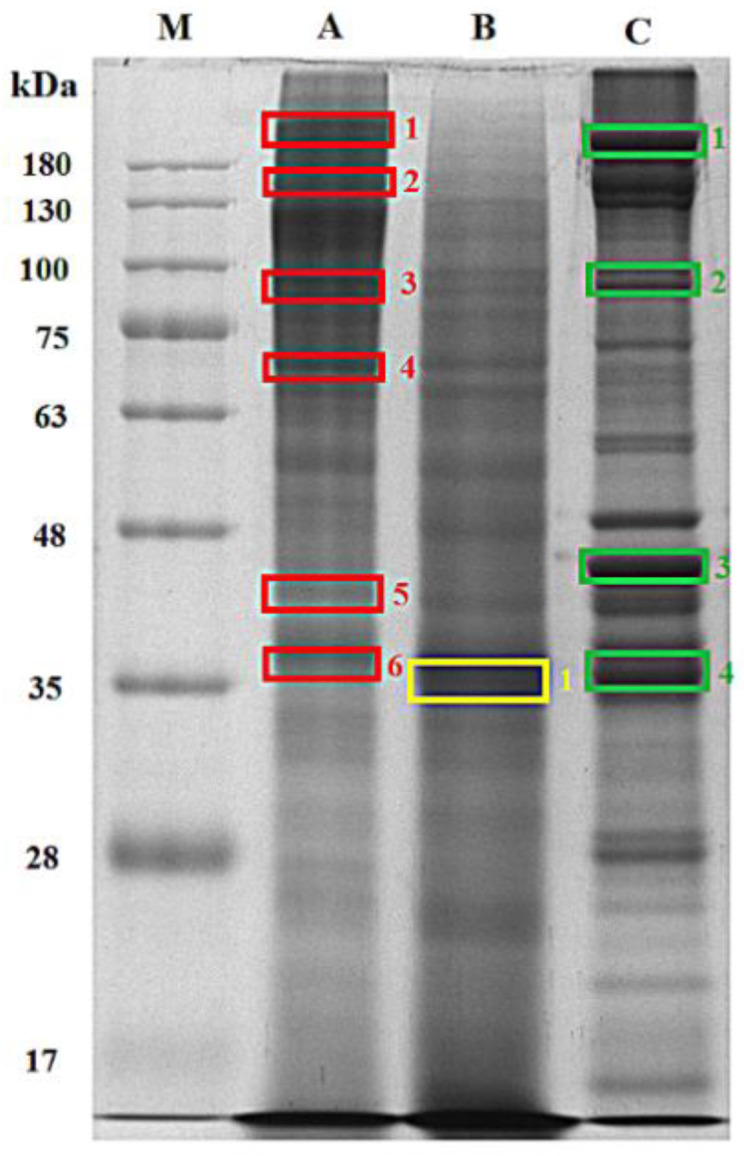
Sodium dodecyl sulfate polyacrylamide gel electrophoresis (SDS-PAGE) of lane A: mackerel steaming juice (MSJ) with defatted process, lane B: mackerel steaming juice with SDS extraction (MSJ SDS), Lane C, mackerel muscle SDS extract; lane M, protein marked using 10% resolving gel and 4% stacking gel.

**Figure 2 foods-11-01785-f002:**
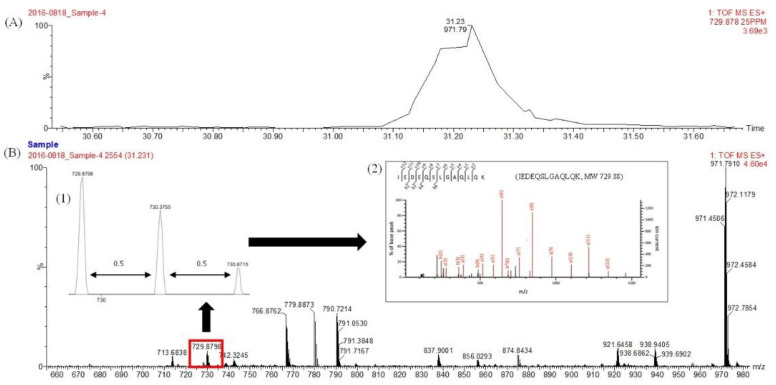
(**A**) LC-MS/MS chromatogram of tryptic peptides identified from MSJ protein band A4 (Figure 1) at 31.24 min. (**B**) The representative spectrum of LC-MS/MS with doubly charged signal (1) and fragmentation spectrum (2).

**Figure 3 foods-11-01785-f003:**
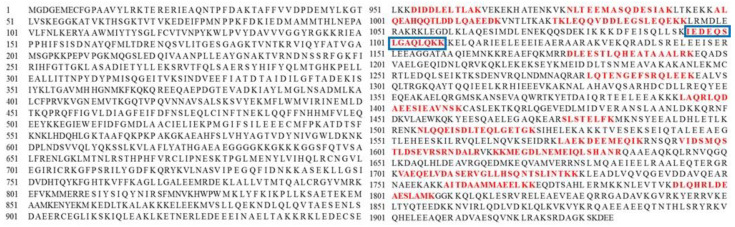
Sequences of myosin heavy chain (fast skeletal muscle; SwissProt: Q90339). Matching tryptic peptides to MSJ band A4 (Figure 1) were marked in red letters.

**Figure 4 foods-11-01785-f004:**
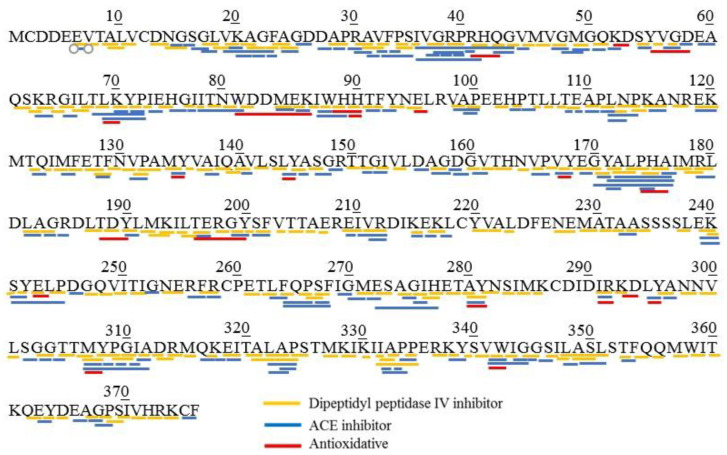
Potential bioactive peptides profiling of actin (SwissProt: P49055) identified from Mackerel Steaming Juice (MSJ).

**Figure 5 foods-11-01785-f005:**
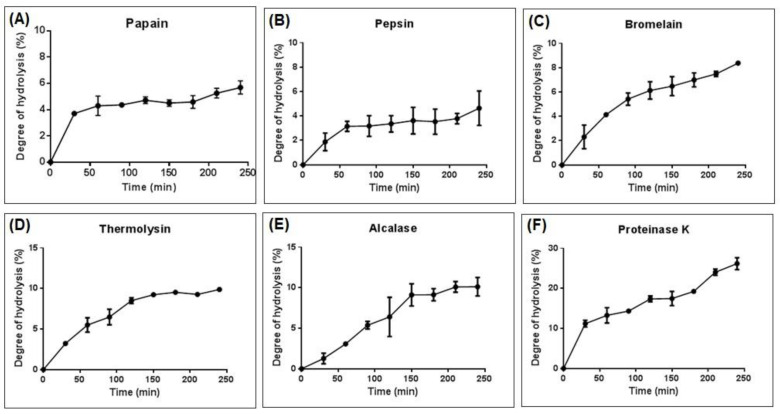
Degree of hydrolysis (%) of mackerel steaming juice (MSJ) powder hydrolyzed for 4 h using (**A**) papain, (**B**) pepsin, (**C**) bromelain, (**D**) thermolysin, (**E**) alcalase and (**F**) proteinase K.

**Figure 6 foods-11-01785-f006:**
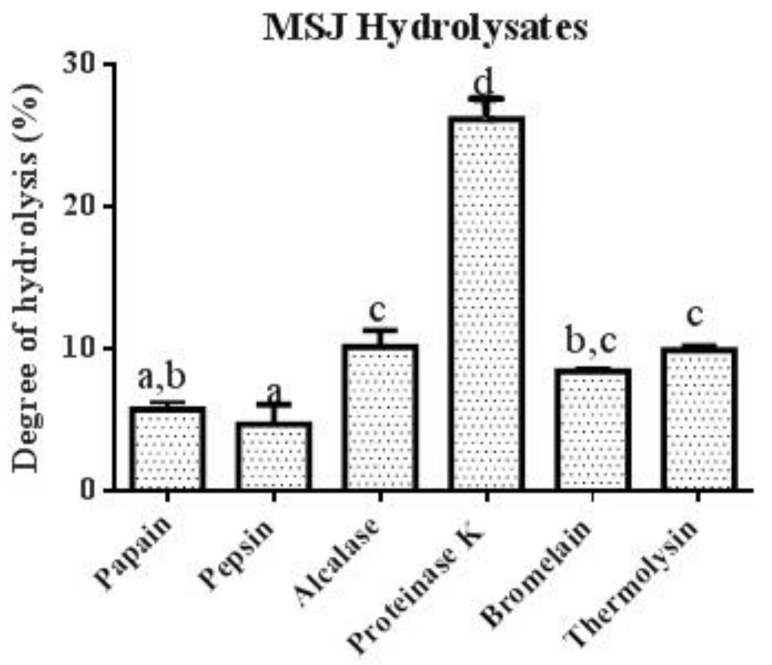
Degree of hydrolysis (%) of MSJ hydrolysates with six proteases at 4 h of hydrolysis. Different alphabets on each bar show significant difference at *p* < 0.05.

**Figure 7 foods-11-01785-f007:**
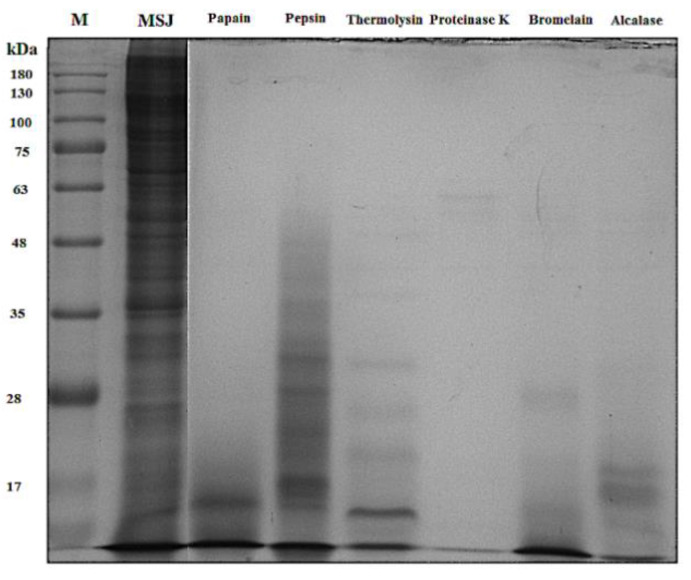
SDS-PAGE protein patterns of Mackerel Steaming Juice (MSJ) and MSJ hydrolysates from six proteases (papain, pepsin, thermolysin, proteinase K, bromelain, and alcalase).

**Figure 8 foods-11-01785-f008:**
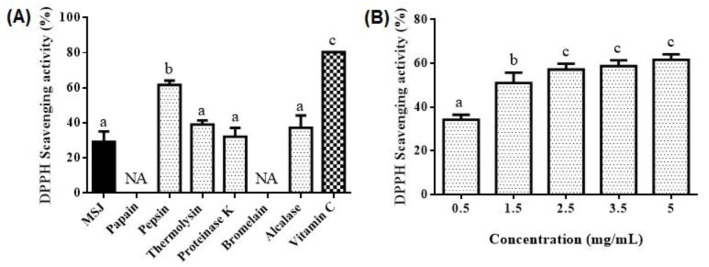
(**A**) DPPH radical scavenging activity of MSJ (5 mg/mL), MSJ hydrolysates (5 mg/mL) and Vitamin C (0.05 mg/mL). (**B**) DPPH radical scavenging activity of pepsin hydrolysates with different concentration. Different alphabets on the bar show significant difference at *p* < 0.05. NA indicates no activity found.

**Figure 9 foods-11-01785-f009:**
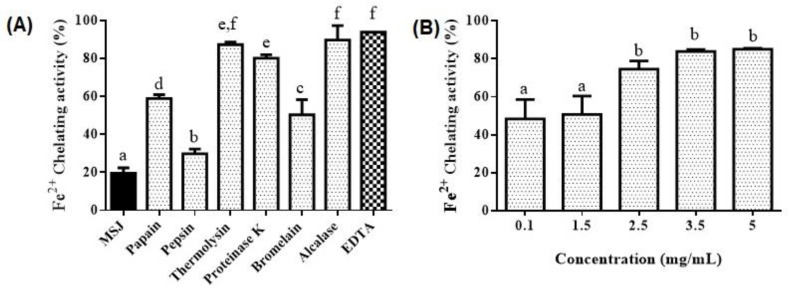
(**A**) Fe^2+^ chelating activity of MSJ, MSJ hydrolysates and EDTA. (**B**) Fe^2+^ chelating activity of alcalase hydrolysates with different concentration. Different alphabets on the bar show significant difference at *p* < 0.05.

**Figure 10 foods-11-01785-f010:**
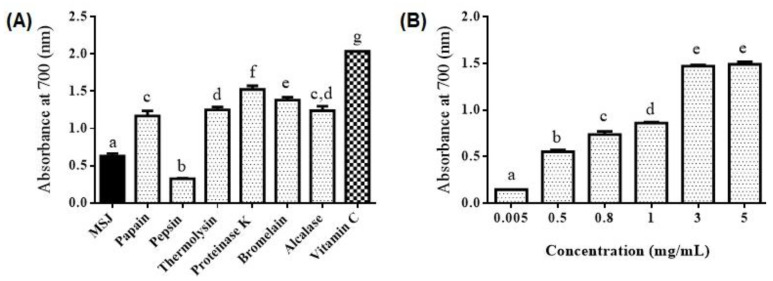
(**A**) Reducing power of mackerel steaming juice (MSJ), MSJ hydrolysates and vitamin C. The concentration of MSJ, MSJ hydrolysates and Vitamin C are at 5, 5, 0.05 mg/mL, respectively. (**B**) Reducing power of proteinase K hydrolysates with different concentration. Different alphabets on the bar show significant difference at *p* < 0.05.

**Figure 11 foods-11-01785-f011:**
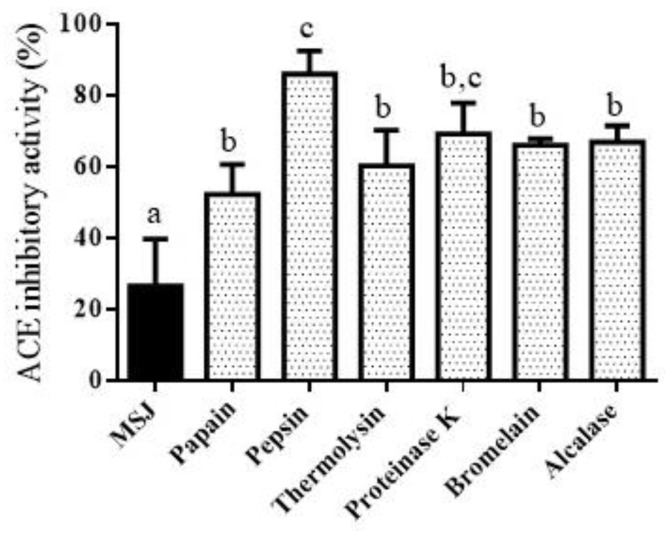
Angiotensin-converting enzyme I (ACE-I) inhibitory activity of mackerel steaming juice (MSJ) and MSJ hydrolysates at 0.1 mg/mL, respectively. Different alphabets on the bar show significant difference at *p* < 0.05.

**Table 1 foods-11-01785-t001:** Proximate composition, NaCl content and pH of mackerel steaming juice (MSJ).

Sample	Moisture (%)	Crude Protein (%)	Crude Fat (%)	Ash (%)	NaCl (%)	pH
Mackerel steaming juice	93.76 ± 0.67	5.30 ± 0.07	0.19 ± 0.01	0.84 ± 0.02	0.38 ± 0.17	5.93 ± 0.08

All data are shown as mean ± SD (*n* = 3).

**Table 2 foods-11-01785-t002:** Summary of proteins identified from mackerel steaming juice (MSJ), MSJ and mackerel meat SDS extract by SDS-PAGE and LC-MS/MS analysis.

Protein	Molecular (kDa) Weight from Database	Molecular Weight (kDa) Estimated from SDS-PAGE		
		**MSJ**	**MSJ** **SDS**	**Muscle** **SDS**
Collagen alpha-2(I) chain	126.908	241 (A1), 127 (A2), 42 (A5)	-	-
Myosin heavy chain, fast skeletal muscle	221.462	91 (A3), 70 (A4),42 (A5), 37 (A6)	37 (B1)	37 (C4), 42 (C3), 242 (C1), 90 (C2)
Actin, alpha skeletal muscle	41.944	42 (A5), 37 (A6)	37 (B1)	37 (C4), 42 (C3), 90 (C2), 242 (C1)
Actin, alpha skeletal muscle B	41.950	-	-	42 (C3)
Actin, cytoplasmic 2	41.726	-	-	90 (C2), 42 (C3)
Actin, alpha anomalous	41.952	-	-	42 (C3), 37 (C4)
Actin, cytoplasmic 3	41.756	-	-	42 (C3), 37 (C4)
Tropomyosin alpha-1 chain	32.767	92 (A3), 70 (A4), 42 (A5), 37 (A6)	37 (B1)	37 (C4), 42 (C3), 242 (C1)
Beta-enolase	47.257	241 (A1), 70 (A4), 42 (A5)	37 (B1)	37 (C4), 42 (C3), 90 (C2), 242 (C1)
Fructose-bisphosphate aldolase A	40.044	241 (A1), 42 (A5), 37 (A6)	37 (B1)	37 (C4), 42 (C3), 90 (C2), 242 (C1)
Glyceraldehyde-3-phosphate dehydrogenase	35.761	241 (A1), 91 (A3),70 (A4), 42 (A5), 37 (A6)	37 (B1)	37 (C4), 42 (C3), 90 (C2), 242 (C1)
Myosin light chain-1, skeletal muscle isoform	20.054	42 (A5)	37 (B1)	37 (C4), 42 (C3), 90 (C2), 242 (C1)

**Table 3 foods-11-01785-t003:** Potential bioactive peptides of identified from mackerel steaming juice using BIOPEP’s Profiles of potential biological activity tool.

Protein	Number of Bioactive Peptides				
	DPP-IV Inhibitor	ACE Inhibitor	Antithrombotic	Antioxidative	Antiamnestic
Collagen alpha-2(I) chain	1056	993	236	49	214
Myosin heavy chain, fast skeletal muscle	1174	657	8	118	4
Actin, alpha skeletal muscle	247	165	3	21	2
Tropomyosin alpha-1 chain	148	85	1	26	0
Beta-enolase	270	185	19	1	0
Fructose-bisphosphate aldolase	237	166	3	25	4
Glyceraldehyde-3-phosphatedehydrogenase	233	152	0	20	2
Myosin light chain-1, skeletal muscle isoform	122	89	0	3	0
Total	3487	2492	270	263	226

**Table 4 foods-11-01785-t004:** Numbers of potential bioactive peptide released from mackerel steaming juice (MSJ) proteins through proteolysis simulation using “enzyme action” tool in BIOPEP-UWM database.

Enzyme	Total number of Bioactive Peptides Released from Mackerel Steaming Juice (MSJ) Proteins *				
	Dipeptidyl Peptidase IV Inhibitor	ACE Inhibitor	Antithrombotic	Antioxidative	Antiamnestic
Ficain	418	275	37	36	35
Papain	296	246	3	23	2
Pepsin (pH > 2)	351	276	31	29	31
Chymotrypsin C	245	188	32	20	32
Proteinase K	362	230	31	27	31
Thermolysin	225	171	0	41	0
Pancreatic elastase	274	184	35	25	35
Cathepsin G	110	78	1	16	0
Bromelain	248	194	24	22	23

Numbers represent the sum of bioactive peptides released from mackerel steaming juice proteins. * MSJ proteins: the combination of collagen alpha-2(I) chain; myosin heavy chain; actin, alpha skeletal muscle; tropomyosin alpha-1 chain; beta-enolase; fructose-bisphosphate aldolase; glyceraldehyde-3-phosphatedehydrogenase; myosin light chain-1.

**Table 5 foods-11-01785-t005:** Peptide contents and yield of mackerel steaming juice (MSJ) hydrolysates.

Sample	Peptide Contents (mg/g)	Yield (%) *
MSJ	162.14 ± 0.001 ^b^	-
MSJ papain hydrolysates	195.21 ± 0.008 ^c^	88.29
MSJ pepsin hydrolysates	124.00 ± 0.003 ^a^	92.79
MSJ thermolysin hydrolysates	195.26 ± 0.008 ^c^	92.79
MSJ proteinase K hydrolysates	326.92 ± 0.007 ^e^	98.20
MSJ bromelain hydrolysates	187.45 ± 0.005 ^c^	89.19
MSJ alcalase hydrolysates	260.05 ± 0.005 ^d^	90.99

Peptide content is shown as mean ± SD (*n* = 3). Different superscripts within the same column indicate the significant differences (*p* < 0.05). * Yield (%) was calculated based on dry final weight over initial weight before hydrolysis.

## Data Availability

The datasets generated for this study are available upon request to the corresponding author.

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
