# Peer review of "In Silico and In Vitro Analyses of Angiotensin-I Converting Enzyme Inhibitory and Antioxidant Activities of Enzymatic Protein Hydrolysates from Taiwan Mackerel (Scomber australasicus) Steaming Juice"

_foods, 2022, doi:10.3390/foods11121785_

Round 1
Reviewer 1 Report
The authors report a new processing step through the use of proteases to unleash the antioxidant potential of mackerel processing by-products. The application is of interest considering the developing field of functional food.
The manuscript presents the experimental approach in detail and is well written.
Fig1- the authors identify the bands that are later selected for protein identification, please add also the corresponding protein by combining with figure 2 and 3. moreover, bands are numbered but these numbers are later not clearly identified in respect to proteins
PAge 11 the title of table 4 is far from the table that needs a more detailed description
table 4 and fig 4 present the same info, please combine
figure 3 can be presented as supplementary files
Reviewer 2 Report
Totally, the study has been well designed. Although, there are some points to review.
Line 40: please add a reference for the first paragraph of introduction.
Line 75 or 78: please use new study that published.
Table 5: Please add standard deviation for each mean.
Fig 11: please double check the statistical analysis.
